# Heat Pump Dryer Design Optimization Algorithm

**Bernardo Andrade [1,2], Ighor Amorim [2,3], Michel Silva [2,3,4], Larysa Savosh [5] and Luís Frölén Ribeiro [2,6,*]**

[1] Centro Federal de Educação Tecnológica de Minas Gerais, Belo Horizonte 30421-169, Brazil; bernardocandrade@hotmail.com

[2] Mechanical Technology Department, Polytechnic Institute of Bragança, 5300-252 Bragança, Portugal; ighor@alunos.utfpr.edu.br (I.A.); michel_rp_15@hotmail.com (M.S.)

[3] Departamento Acadadêmico de Mecânica, Universidade Tecnológica Federal do Paraná, Ponta Grossa 84017-220, Brazil

[4] Team4cooling, 2795-898 Terrugem, Portugal

[5] Department of International Economic Relations, Lutsk National Technical University, 43000 Lutsk, Ukraine; larysa.savosh9@gmail.com

[6] Centre for Renewable Energy Research—INEGI, 4200-465 Porto, Portugal

\* Correspondence: frolen@ipb.pt; Tel.: +351-27-3303148

**Abstract:** Drying food involves complex physical atmospheric mechanisms with non-linear relations from the air-food interactions, and those relations are strongly dependent on the moisture contents and the type of food. Such dependence makes it complex to design suitable dryers dedicated to a single drying process. To streamline the design of a novel compact food-drying machine, a heat pump dryer component design optimization algorithm was developed as a subprogram of a Computer Aided Engineering tool. The algorithm requires inputting food and air properties, the volume of the drying container, and the technical specifications of the heat pump off-the-shelf components. The heat required to dehumidify the food supplied by the heat exchange process from condenser to evaporator, and the compressor's requirements (refrigerant mass flow rate and operating pressures) are then calculated. Compressors can then be selected based on the volume and type of food to be dried. The algorithm is shown via a flow chart to guide the user through three different stages: Changes in drying air properties, heat flow within dryer and product moisture content. Example results of how different compressors are selected for different types of produces and quantities (*Agaricus blazei* mushroom with three different moisture contents or fish from *Thunnini* tribe) conclude this article.

**Keywords:** algorithm; heat-pump; drying; food; design; optimization

## 1. Introduction

Food drying is one of the strategies for food preservation, and one of several strategies is the thermal based mechanism, which is complex and involves the removal of a solid product's moisture by the employment of heat. The drying occurs through heat and mass transfers while the properties of the food change throughout the process. There are many different machines that can make this process, one of these is heat-pump based [1].

This heat pump based machine extracts energy via a gas compressor from a cold-source and delivers it to a hot-source. It does so by providing work to the refrigerant fluid. The heat-pump provides heat, making it directly useful for heating, ventilation and air conditioning (HVAC), but also for drying applications. The machine energy input is the energy received from the compressor and that is added to the amount of energy removed from the cold-source, yielding a higher energy output as heat. As an example, for a compressor yielding 100 W to remove 400 W from a cold-source,

the total amount of energy provided to the hot-source will be 500 W. This is a 5-time higher value than the one extracted by the compressor, meaning a 500 W heating service from a 100 W electrical input. This highlights the energy saving feature of this technology. A scheme of the whole machine and its components is shown in Figure 1: Compressor, expansion valve and heat exchangers (condenser and evaporator).

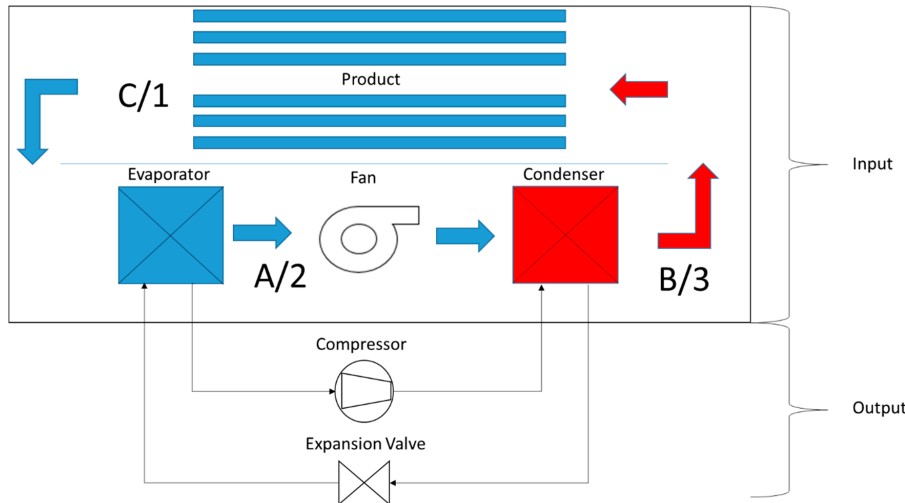

**Figure 1.** Scheme of a specific prototype of a heat pump food dryer tested for this algorithm where the thin line represents the refrigerant circuit while the thick arrows represent the airflow circuit.

The work cycle starts with the air being heated at the condenser after being blown towards the product (A to B in Figure 1). When the warm air passes by the food, it removes moisture from the material. Downstream, the air captured the moisture from the food (B to C in Figure 1) and proceeds to the evaporator, to partially condense some of its water. This (C to A in Figure 1) is achieved by promoting the heat transfer of the air with the surfaces of the evaporator colder than the dew-point of the air. For a real machine, a percentage of outside air replaces part of the circulating air at each pass to allow the cooling elements to condense more moisture from the air and to avoid the increase in temperature of the circulating air.

The design of this machine for different purposes, or types of food, can be improved by the application of resources such as the numerical optimization. Improving equipment design takes time and laboratorial costs which are not directly translated towards the manufacturing process, for the material acquired is usually spent on tests. Mathematical and computational mechanics models are effective alternatives to many practical experiments since they provide a prediction of what may happen and what can be expected. This approach allows for greater comprehension of the transport phenomena involved in the drying of food, sharpens testing and production leading to a better and quicker design process [2].

A simulation can also have an edge on normal experiments for it can predict, with virtual sensors, the humidity, air velocity and temperature on points that are normally inaccessible since the presence of the sensors would influence the air circulation inside the drying container. In addition, there is no limitation of testing different working conditions, there is no space restriction nor need for trained operators. However, simulations require reliable data and proper modelling, otherwise the quality of results may be questionable. This is true for the case of drying food especially describing the physical/chemical properties and transport phenomena.

To achieve better results in energy efficiency and product quality, studies have been performed in this area and new, more efficient models of heat pump dryers have been created [3,4]. Thus, the use of algorithms comes into play as the driving force of better product design, enabling the creation of products directed to many markets: from food industry to the small-scale farmer.

An algorithm is proposed in this article to aid the selection of heat pump components. It is presented through a flow chart that helps the designer visualize and comprehend the specified numerical solution's steps. A code was created to test the algorithm and how its results help the development of different size drying machines. Examples from two different produces and how they determine compressor selection (outputs) are later discussed.

## 2. Materials and Methods

The design algorithm is a logic map of the efficient design of food-drying heat pump air-based machines. The algorithm inputs are the type of food, the air properties and the dimensions of the food container, as well as the dryer component properties. To guarantee the temperature control throughout the drying process, the working temperature of the air to which the food is being exposed to is used as input. Therefore, it is possible to ascertain the final product quality, since over-heating could cause damage to the product. Meanwhile, the algorithm outputs the mass flow of refrigerant fluid from which the compressor and expansion valve can be selected. The difference of the algorithm inputs and outputs, and their relation to the machine design are depicted in Figure 1. In this figure within the circulating air volume there are 2 different nomenclatures aiming the description of different things: A, B and C are related only to the air properties; and 1, 2 and 3 represent the 3 stages of the algorithm, with different transport and energy equations:

1.  Changes in drying air properties;
2.  Heat flow within dryer;
3.  Product moisture content.

### 2.1. Stage 1 of the Algorithm

The first stage of the algorithm determines the psychrometric and dynamic states of the air. The literature recommends temperatures for drying each food, and by using them in the following psychrometric and transport equations, one obtains every property necessary to the characterization of the process and posterior steps [5–7].

The calculus of the air properties are given by Equations (1)–(17). The specific psychrometry Equations (1)–(8) are recommended by [6]. The psychrometric equations utilized at this stage describe the humid air based on its temperature, amount of water as vapor and the air's occupied volume [8–10].

To provide the readers less clutter information on the variables of the equations, a list of variables and units is shown at the end of the article.

At first it is necessary to obtain the vapor saturation pressure $P_{vs}$ and the absolute humidity $w$. The air humidity after the contact with the food in the first iteration and after the air leaves the evaporator, corresponding to air process C and A in Figure 1, are calculated by Equations (1) and (2):

$$P_{vs} = 6 \times \frac{10^{25}}{1000 \times T^5} \times exp\left(-\frac{6800}{T}\right) \tag{1}$$

$$w = \frac{0.622\, P_v}{P_{atm} - P_v} \tag{2}$$

In the following iterations, the absolute humidity is obtained by adding the total humidity lost by the food to the air humidity after it left the condenser, process B to C in Figure 1. The air humidity when it leaves the condenser is equal to the one when it left the evaporator. Therefore, the next step are the calculations of the air's vapor pressure $P_v$, and its enthalpy $H$, from the absolute humidity Equations (3) and (4):

$$P_v = w \times \frac{P_{atm}}{0.622 + 0.378 \times w} \tag{3}$$

$$H = 1.006 \times (T - 273.15) + w[2501 + 1.775 \times (T - 273.15)] \tag{4}$$

With the air's vapor pressure and the vapor saturation pressure, the relative humidity $\varnothing$ and the dew point $Tdp$ are then calculated for this pressure, Equations (5) and (6).

$$\varnothing = \frac{P_v}{P_{vs}} \tag{5}$$

$$T_{dp} = \frac{186.4905 - 237.3 \log 10 P_v}{\log(10 P_v) - 8.2859} \tag{6}$$

Additionally, with the vapor's pressure and the absolute humidity, the algorithm calculates the specific volume $v$ and the vapor molar fraction $x_v$ relative to the mixture and molar mass.

$$x_v = \frac{P_v}{P_{atm}} \tag{7}$$

$$v = 0.28705T \times \frac{1 + 1.6078w}{P_{atm}} \tag{8}$$

The determination of the transport properties, Equations (11)–(17) as stated by [7], require the non-dimensional dry air/water vapor proportion parameters $\Phi_{av}$ and $\Phi_{va}$, Equations (9) and (10) also recommended by [7].

$$\Phi_{av} = \frac{\sqrt{2}}{4}\left(1 + \frac{M_a}{M_v}\right)^{-\frac{1}{2}}\left[1 + \left(\frac{\mu_a}{\mu_v}\right)^{\frac{1}{2}}\left(\frac{M_a}{M_v}\right)^{\frac{1}{4}}\right]^2 \tag{9}$$

$$\Phi_{va} = \frac{\sqrt{2}}{4}\left(1 + \frac{M_v}{M_a}\right)^{-\frac{1}{2}}\left[1 + \left(\frac{\mu_v}{\mu_a}\right)^{\frac{1}{2}}\left(\frac{M_a}{M_v}\right)^{\frac{1}{4}}\right]^2 \tag{10}$$

The acronyms $M_v$ and $M_a$ respectively represent the molar masses of the vapor and dry air while the $\mu_v$ and $\mu_a$ represent their dynamic viscosity. Therefore, with the proportion parameters $\Phi_{av}$ and $\Phi_{va}$ defined, the mean thermophysical properties are calculated.

Firstly, the thermal conductivity of the humid air $k_{air}$ is given by Equation (11):

$$k_{air} = \frac{(1 - x_v)k_a}{(1 - x_v) + x_v\Phi_{av}} + \frac{x_v k_v}{x_v + (1 - x_v)\Phi_{av}} \tag{11}$$

The specific heat $c_{pm}$ of this air is obtained with Equation (12).

$$c_{pm} = c_{pa}x_a\frac{M_a}{M_m} + c_{pv}x_v\frac{M_v}{M_m} \tag{12}$$

These properties are used to obtain the thermal diffusivity $\alpha$, which is expressed by Equation (13):

$$\alpha = \frac{k}{\rho \times C_{pm}} \tag{13}$$

Additionally, the mixture density $\rho$ for incompressible gases are calculated according to Equation (14):

$$\rho = \frac{P_0}{RT}\left[1 - x_v\left(1 - \frac{M_v}{a}\right)\right] \tag{14}$$

The transport properties that govern the fluid's movement are calculated from the equations pointed by [7]. The dynamic viscosity $\mu_{mix}$ of the mixture results from Equation (15):

$$\mu_{mix} = \frac{(1 - x_v)\mu_{air}}{(1 - x_v) + x_v\Phi_{av}} + \frac{x_v\mu_{vapor}}{(1 - x_v) + x_v\Phi_{va}} \tag{15}$$

With $\mu_{mix}$ and $\rho$, the cinematic viscosity $\tau$ then is obtained with Equation (16):

$$\tau = \frac{\mu_{mix}}{\rho} \tag{16}$$

The Prandl number *Pr*, which is used to determine the water loss, is calculated from Equation (17):

$$Pr = \mu_{mix}\frac{c_{pm}}{k_{air}} \tag{17}$$

*2.2. Stage 2 of the Algorithm*

The second stage of the algorithm relates to the heat flow analysis and component design, it uses the data calculated in Stage 1, the pre-determined dimensions and construction parameters of components to calculate results that are essential to the final design of the product.

To reduce the time from client order to the actual manufacturing of the novel compact food-drying machine, a supplier component database is created from which off-the-shelf products such as heat exchangers and fans will be selected. Their physical dimensions and operating parameters are used in the algorithm. As a heat pump system can be defined by the compressor and the heat exchangers [9], the heat output of this stage will be used to calculate the mass flow rate of refrigerant required to select a fitting compressor.

The reasoning behind this is that controlled temperatures are a main focus of the algorithm to assure product quality. Additionally, the heat exchangers and fans are parts that affect the final product dimension if changed, and so, by selecting products that are available in the market, the cost of production is expected to drop and the final product construction can be streamlined. Any machine designed through this method will have its power output controlled through the variation of the compressor's cycle rate.

At this stage, the fan's diameter is used with previously calculated air speed and density to obtain the air mass flow rate, which will be used in the third stage for controlling the removed water.

With the combined data from Stage 1 and the dimensions of components, the heat which will flow to the air is calculated. That heat is the same that is removed from the refrigerant fluid, and since the temperatures have been set, the enthalpy variation of the refrigerant expected is known and so its mass flow rate is achieved.

To do so, the equations used were the ones that relate to the heat exchangers, such as logarithmic mean temperature difference, Nusselt and Reynolds dimensionless numbers, global heat conductivity and heat flow equation at heat exchangers. These equations are pointed out in [10–13].

The logarithmic mean temperature difference $\Delta T_{ml}$ is a variable that accounts for the logarithmic nature of the heat transfer properties and converts the temperatures and the exchanger's entry and exit, jointly with the external's fluid temperature to obtain a mean value that can be used in heat transfer equations.

$$\Delta T_{ml} = \frac{(T_{med} - T_{in}) - (T_{med} - T_{out})}{\log \frac{(T_{med} - T_{in})}{(T_{med} - T_{out})}} \tag{18}$$

These equations also require a mean global heat flux coefficient *U*. This has the same principle of the previous Equation (18), making a mean value that accounts for every heat transfer process. However, unlike the logarithmic mean temperature difference, this equation results in a heat transfer factor.

$$U = \frac{1}{\frac{1}{h} + \frac{l}{k}} \tag{19}$$

where *l* represents the thickness of the heat exchanger's walls.

Therefore, the required data to calculate such a factor are the conduction heat transfer coefficient for the heat exchanger's material $k$, and the convection heat transfer coefficient for the operating air flow $h$.

$$h = k_{air} \frac{Nu}{D} \tag{20}$$

where D equals to the heat exchanger's cylinder diameter $k_{air}$ is the air's heat conductivity and $Nu$ is a dimensionless number obtained through the following equation:

$$Nu = 1.13\, Re^M \times C \times Pr \tag{21}$$

In this equation, the $M$ and $C$ variables are constants obtained based on the heat exchanger's dimensions and layout. The $Re$ is another number obtained by:

$$Re = \frac{\rho V D}{\mu} \tag{22}$$

The $V$ in the equation is the air's speed. Finally, the final exchanged heat value $\dot{Q}$, equals to:

$$\dot{Q} = UA\Delta T_{ml} \tag{23}$$

where $A$ is the total exposed heat exchanger area. The heat can be used, as previously mentioned, together with the variation of enthalpy $\Delta H$, to obtain the refrigerant mass flow rate $\dot{m}$.

$$\dot{m} = \frac{\dot{Q}}{\Delta H} \tag{24}$$

*2.3. Stage 3 of the Algorithm*

The third stage, the food analysis, is a control stage. This means that in it the algorithm has its control variables calculated to provide the iterative results that make the calculating cycles continue or stop. For this case, the control variable is food's moisture level, and it is calculated through the use of well-known food-drying models. The Modified Henderson model was selected for its recurring appearance in the literature and consequent versatility. It requires the air's humidity level, temperature and speed to calculate the water loss variation [4,5].

For the calculus of the water mass transfer and consequently, total moisture left in the product, the air diffusion coefficient is used as cited at [11,14]. This coefficient is presented in Equation (25):

$$D_{ab} = 1.87 \times 10^{-10} \frac{T^{2.072}}{P} \tag{25}$$

This equation will lead to an underestimation of the drying time for it does not consider the biological properties of internal moisture diffusion and surface diffusion. However, the algorithm structure is built to incorporate further published knowledge in this particular field.

With the air diffusion coefficient calculated, the Graschof $Gr$ and Schimdt $Sc$ numbers can be obtained:

$$Gr = \frac{g\Delta\rho S^3}{\rho\tau^2} \tag{26}$$

$$Sc = \frac{\tau}{D_{ab}} \tag{27}$$

where $S$ is the characteristic dimension, which in the case of the drying machine are the spaces between the plaques that hold the food. With both Graschof and Schimdt, the Rayleigh $Ra$ and Sherwood numbers can be obtained, for both natural $Sh_n$ and forced $Sh_f$ convections:

$$Ra = Gr \times Sc \tag{28}$$

$$Sh_n = 0.197 \times Ra^{\frac{1}{4}} \left( \frac{hp}{S} \right)^{\frac{1}{9}} \tag{29}$$

If Reynolds is less than 200.000,

$$Sh_f = 0.664 \times Re^{0.5} \times Sc^{\frac{1}{3}} \tag{30}$$

However for value greater than that,

$$Sh_f = 0.0365 \times Re^{0.8} \times Sc^{\frac{1}{3}} \tag{31}$$

With Sherwood defined, the mass transfer coefficient is obtained with Equation (32).

$$hcf = Sh\frac{D_{ab}}{hp} \tag{32}$$

The *hp* value is the food containing plaque's height. The total water mass removed $m_l$ is calculated by:

$$m_l = hcf \times ns \times Ap \times \Delta\rho \tag{33}$$

In Equation (33), *Ap* is the plaque's area, *ns* is the number of plaques and $\Delta\rho$ is the difference between density of water in the air and food.

The next step of the algorithm compares the value obtained with the mass transfer equation and the Modified Henderson model, and select the most conservative value. This value is removed from the food's total humidity and accounted for in the control function, restarting the cycle if necessary.

## 3. Results

The algorithm can be represented in the form of a flow chart to illustrate the proposed logic map. The first stage shown in Figure 2 results in the definition of the air's psychrometric and transport properties throughout the drying process, doing so from Equations (1)–(17), and the input data both from the food and from the machine.

The first decision box of the flowchart present in this first stage is a part of the logical process of the algorithm and should be included in any code as failsafe. The halting of the calculation process is given when the variation of the moisture content over time reaches almost zero, 1E-9. This criterion is flexible because it accommodates residual moisture differences from different types of food.

The properties outputs are used mainly as input for other stages. However, the program still can provide this data to guarantee quality control, specifically through the monitoring of the air's temperature at the exit of each component.

The second stage shown in Figure 3 outputs both design and process parameters. By taking the aforementioned properties, it calculates the heat required to change the air's state at both condenser and evaporator. With it, and the expected enthalpy variation, the refrigerant mass flow rate is achieved. These characteristics allow for an easy selection of the components required to design the heat pump of the dryer. These components are: Compressor, expansion-valve, refrigerant fluid and heat exchangers.

Even though the algorithm allows for easier selection of components, there are still some parts that require manual selection. As commented in Section 2, the fans that circulate the air are pre-selected to fit the drying container, so that their dimensions are used as input data to calculate the mass flow rate of the circulating air inside the machine.

For the final stage shown in Figure 4, the results are given as a function of the amount of water in the system. The algorithm outputs the rate of water removal. From it, the algorithm calculates how

this rate varies and how it effects the drying food. Finally, the variation of how much water is being removed is used as a parameter for cycle control and break function.

The whole algorithm is depicted in Figure 5 showing the 3 stages that correspond, in Figure 1, to the A–B, B–C and C–A thermal processes.

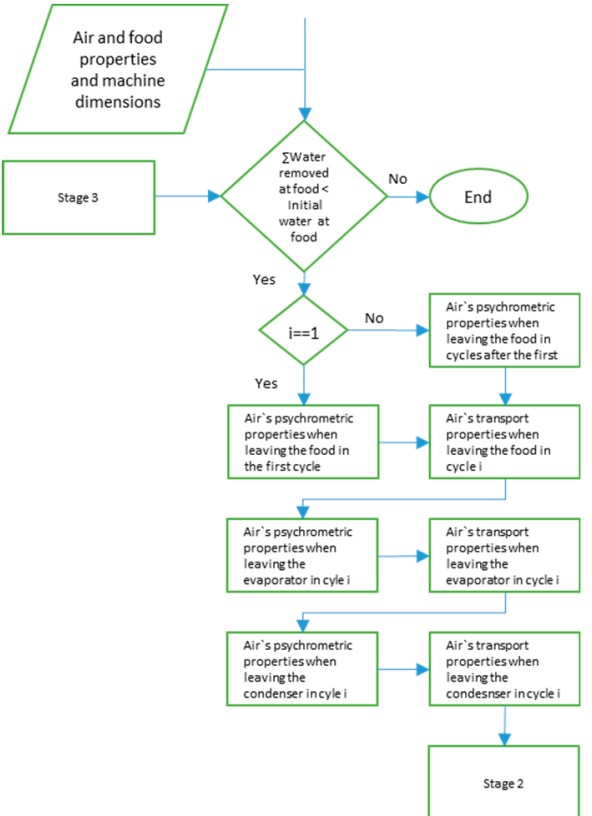

**Figure 2.** Stage 1 of the algorithm. The internal air analysis described in Equations (1)–(17).

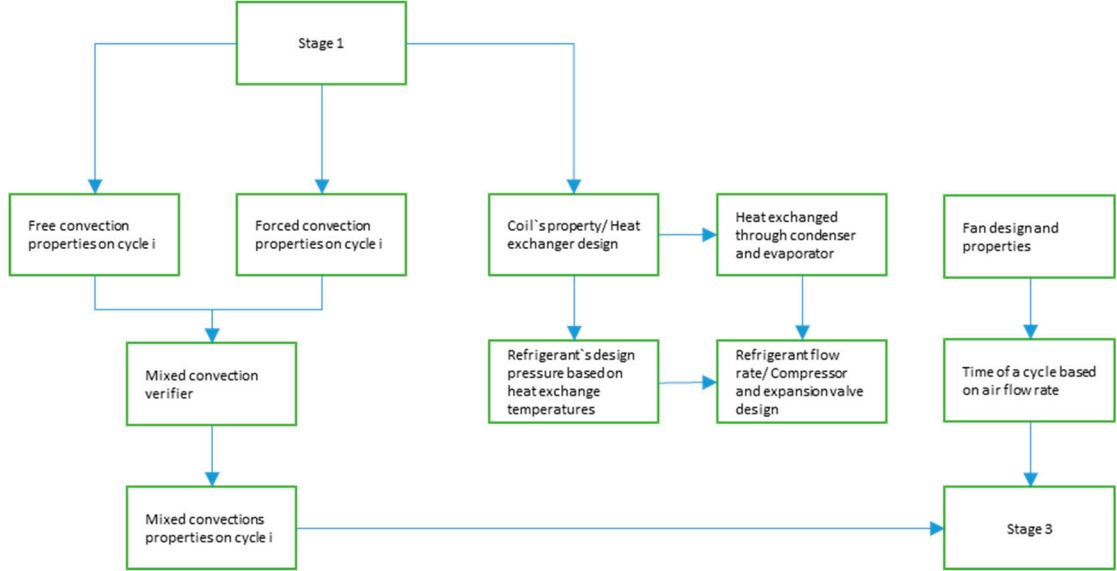

**Figure 3.** Stage 2 of algorithm. Heat flow analysis between components, air and food. Component design derived from Equations (18)–(24).

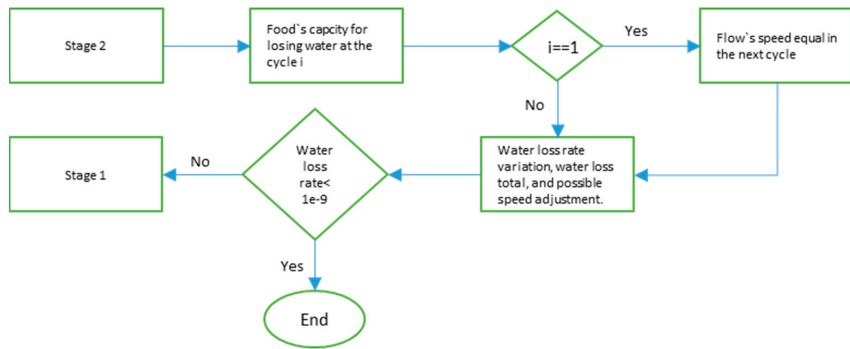

**Figure 4.** Stage 3 of algorithm. Food humidity calculus and verification as demonstrated from Equations (25)–(33).

**Figure 5.** Heat pump dryer design optimization algorithm

## 4. Discussion

The flow chart in Figure 5 allows for an easier overview of the process. This handy resource guides its user from a basic starting point towards its desired goal which makes designing simpler since every information required can be traced back.

The proposed algorithm is a tool of how to design the heat pump air-based drier and is also a step-by-step guide. The value proposition is by providing the specifications of the components for the machine that is going to be built (compressors, evaporators, condensers, etc.); and the fact that it is not restricted by the food-drying physical models hereby proposed. The latter is a feature of the algorithm because it is possible to update the used models of each process to the newest and most sound ones available. Doing so, and using more accurate data, will affect the precision of the final result. Actually, such a practice was used in the development of this algorithm. Data and equations found in earlier versions of established guides and books such as [8] and [10] were posteriorly replaced by newer ones [7].

The algorithm specifies each step and allows for the comprehension of the necessary and produced data for that step. A code was written in GNU Octave automating the algorithm to deliver as output a final value, and not clutter the user with processual information. In order to exemplify the application of the code, one simulated the drying of the *Agaricus blazei* mushroom for batches of varying volumes that correspond to about 45, 123, 200, 277 and 355 kg of product. To simulate the drying, and consequently provide suitable data for the design, it was considered that the properties pointed out by [5] related to the product's fraction of water. Additionally, the input for stage 2 related to the heat-exchangers and fan dimensions were based on the ECO coils and coolers of the Luvata Company.

Considering that 90% of the drying container's volume is filled with food and that the mushroom's amount of water for three different cases is: 60%, 80% and 88%, the humidity removed can be calculated based on the container's volume. This is used as a variable parameter from which the power output can be measured. Additionally, the temperature of drying was set to 80 °C, the superior temperature limit from which this particular mushroom species starts having chemical and organoleptic changes.

The algorithm simulated moisture removal for those given conditions and the results are depicted in Figure 6. The refrigerant mass flow rate is plotted against the volume of food in the drying container, being a critical information for the selection of a suitable compressor. A commercial compressor for nominal power operations yields a maximum mass flow rate. The graph represents these working limits of three types of compressors from a given manufacturer (EMBRACO). The number I, II and III respectively represent increasingly potent compressors of 610, 990 and 1445 W.

The simulation results show a clear need to upgrade the compressor from type I to II while increasing the amount of *Agaricus blazei* mushrooms from 45 to 123 kg, a 2.73-fold increase in drying needs to a 1.6 increase in compressor power. Furthermore, an increase of 1.6 times, from 123 to 200 kg, will require a power increase of 1.46 times. The non-linearity of the drying process is clear, and in this example, for the larger batches, from 200 to 355 kg, only one type of compressor would suffice.

For smaller batches, the amount of humidity in the food is negligible for design purposes. For larger batches, the power needs are quite different. The dryer the food is, the harder the compressor will have to work as it is depicted in Figure 6. This is a known result and a good indication of the proper response of the algorithm.

The power needs for drying other types of products were also evaluated. Additional simulations were done for fish from the *Thunnini* tribe, comprising 15 species of the vulgarly known as tuna fish. As those species of fish (when fresh) are 3.3 times denser than mushrooms and have constant moisture, more fish will fit into the drying chamber. There are also significant differences between the experimental values of the drying curve from both products [15]. The experimental values for the drying process are seldom available and changes may occur according to how the produce is presented (whole, sliced, filleted, grinded, etc.). For example, a reduction of 1.57 times compressor mass-flow rate (and the same amount in compressor power) is required when drying sliced fresh tuna fish with a moisture content of 80% against the same volume of mushrooms. This contra intuitive result derives

from the fact that the drying temperature of the fish from the *Thunnini* tribe cannot exceed 40 °C, thus a gentler drying is required. Different results may occur if the product is considered in a different presentation, but the lack of public experimental drying curves is a caveat within this industry.

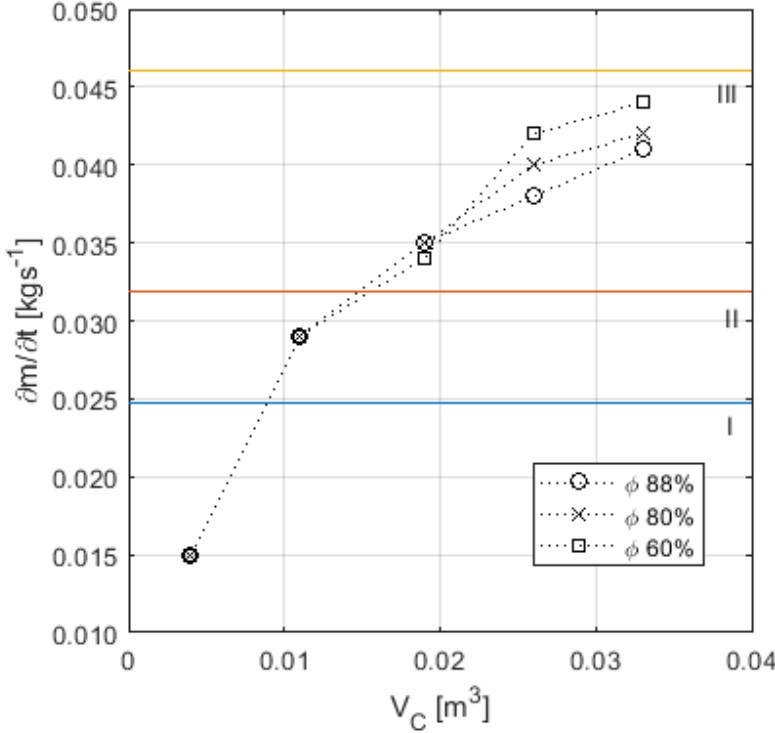

**Figure 6.** Relation of volume of product to required refrigerant flow.

## 5. Conclusions

A flexible optimization algorithm is presented, aimed to help design heat pump air-based dryers incorporating off-the-shelf components. The algorithm is segmented into three parts allowing the modification or upgrade of any one according to new scientific developments. It also allows dedicated solutions for different types and quantities of food because it incorporates their own chemical and organoleptic limitations.

With this guide in hand, the selection of components and materials is simpler because the users will have the key parameters of the required components and streamline the iterative process of machine design.

The authors recommend that future versions of the algorithm, and possible programs, incorporate more precise equations that consider the falling rate diffusion coefficient for the moisture, possibly replacing Equation (25) by a better one or using specific equations for different types of food.

**Author Contributions:** Conceptualization, B.A., I.A., M.S. and L.F.R.; Formal analysis, L.S.; Funding acquisition, L.F.R.; Investigation, B.A., I.A. and M.S.; Methodology, L.S.; Project administration, L.F.R.; Supervision, L.F.R.; Validation, L.S.; Writing–original draft, B.A. and I.A.; Writing–review & editing, L.F.R.

**Funding:** This project was funded by the Polytechnic Institute of Bragança, the Centre for Renewable Energy Research, Portugal, and the Lutsk National Technical University of Ukraine in equal parts.

**Conflicts of Interest:** The authors declare no conflict of interest

## Nomenclature

Roman characters

| | | |
|---|---|---|
| $A$ | Total exposed heat exchanger area | [m$^2$] |
| $H$ | Enthalpy | [kJ/kg * K] |
| $h$ | Convection heat transfer coefficient | [W/m$^2$ * K] |
| $k$ | Heat transfer coefficient of material | [W/m$^2$ * K] |
| $l$ | Thickness of the heat exchanger's walls | [m] |
| $T$ | Absolute temperature | [K] |
| $U$ | Mean global heat flux coefficient | [W/m$^2$ * k] |
| $V$ | Air's speed | [m/s] |
| $w$ | Absolute humidity | [kg water/kg air] |

Greek characters

| | | |
|---|---|---|
| $\alpha$ | Thermal diffusivity | [m$^2$/s] |
| $\rho$ | Mixture density | [kg/m$^3$] |
| $\tau$ | Kinematic viscosity | [m$^2$/s] |
| $v$ | Specific volume | [m$^3$/kg] |
| $\varnothing$ | Relative humidity | |

Subscripts

| | | |
|---|---|---|
| $Ap$ | Plaque's area | [m$^2$] |
| $c_p$ | Specific heat | [kJ/kg * K] |
| $D_{ab}$ | Air diffusion coefficient | [m$^2$/s] |
| $k_{ar}$ | Thermal conductivity of the humid air | [W/m$^2$ * K] |
| $m_l$ | Total water mass removed | [kg/s] |
| $\dot{m}$ or $\frac{\partial m}{\partial t}$ | Mass flow rate | [kg/s] |
| $\mu_{mix}$ | Dynamic viscosity | [N * s/m$^2$] |
| $Pv$ | Air's vapor pressure | [Pa] |
| $P_{vs}$ | Vapor saturation pressure | [Pa] |
| $T_{dp}$ | Dew point | [K] |
| $\Delta T_{ml}$ | Logarithmic mean temperature difference | [K] |
| $x_v$ | Vapor molar fraction | |

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
