# Peer review of "Heat Pump Dryer Design Optimization Algorithm"

_inventions, doi:10.3390/inventions4040063_

Round 1

Reviewer 1 Report

it is better / improved/ than the previous version

you completed most of the requirements I expressed, even in an original form, I would have expected more details

the answers sent to my suggestions are fine and taken into consideration

Author Response

Dear Reviewer 1,

A general check was performed. Math and text typos were identified in the following lines and equations and are highlighted in a dedicated document: 

38, 39, 40, 58, 71, 105,124,132, 179, 270, 303, 332, 333, 343, 349, 357, 358, 361

Eq: 7, 11, 12, 13, 17, 20

Thank you for your inputs in this process!

Reviewer 2 Report

The paper introduces an algorithm to help design heat-pump air-based dryers incorporating off-the-shelf component. The paper is interesting and well presented. In my opinion it could be accepted after the following minor remark will be addressed:

Do not report the list of variables in Appendix A but it is more correct to name this section as Nomenclature. Then all the symbols used in the text should be reported ordered alphabetically and divided in Roman, Greek, Subscripts and Acronyms. Change [pa] in [Pa]. A minor check of the General English is required also to correct some typos.

Author Response

Dear Reviewer 2,

Your recommendation was implemented in the manuscript:

A general check was performed. Math and text typos were identified in the following lines and equations and are highlighted in a dedicated document: 

38, 39, 40, 58, 71, 105,124,132, 179, 270, 303, 332, 333, 343, 349, 357, 358, 361

Eq: 7, 11, 12, 13, 17, 20

Thank you for your inputs in this process!

This manuscript is a resubmission of an earlier submission. The following is a list of the peer review reports and author responses from that submission.

Round 1

Reviewer 1 Report

1. Abstract

Line 14. Delete “Moreover” Suggest join first two sentences with “and”.

Line 15 Moisture content (singular)

Line 16 Change “machines” to “dryers”

Line 16 Delete “speed-up”

Line 17 Delete “proposed”

Line 18 Change “inputs” to “requires inputting”

Line 18 Misspelt “properties”

Line 19 Change “equals” to “is supplied by”

Line 22 “selected”

Line 23 Change to “to guide the user through 3 different stages:” (delete remaining clause)

Line 24 Suggest stages be called:

Changes in drying air properties

Heat flow within dryer

Product moisture content

Line 25 Briefer:

 Results of the algorithm are presented for mushroom (Agaricus Blazei) at 60, 80 and 88% moisture content, and for batches of about 45, 123, 200, 277 and 355 kilograms.

(No need to present detailed results – anyway the results actually presented in the abstract are not correct as they don’t match the treatment variables).

Overall: abstract is longer than necessary, and reading it suggests that the algorithm was only developed for the mushroom experiments, so that the algorithm has not been further tested than for one product.

2. Introduction

Extensive grammatical editing required.

Suggest replacing the word “humidity” with “moisture content: for the product, reserving the word humidity for the air properties only (as is conventional in drying work.

Plan of introduction is good, with quick transition from dryers in general to mechanism of heat pump dryer.

Description of heat pump drying process is rather poor, written in a non-technical way and uses colloquialisms, such as “It then heads to …”

The diagram suggests all of the air is cooled and reheated, but in practise only a small percentage of the total recirculated air should be heat-pump processed with each pass, allowing the cooling elements to condense more moisture from the air. Also the fan is normally located away from the heat exchangers, not between them. Is this a specific design? It purports to be a general design.

Line 75,76: Dryer optimisation is normally concerned with energy efficiency, not new product development.

Apart from this, the introduction provides a suitable background and purpose for the paper.

3. Materials and Methods

Paragraph from lines 83-96 would probably be better placed in the preceding section (Introduction).

Extensive editing of grammar required.

Source (reference) required for equation (1).

The symbol T (not in final nomenclature) is probably absolute temperature.

Line 116 Equation 4 appears to use the wrong value for the specific heat of air (should be about 2.0, some textbooks use 1.996 kJ/kg.K).

Equation for calculating Pvs is missing. Suggest a reference to a standard saturation vapour pressure equation would suffice.

Equations 9-12 are not referenced. Generally if the equation is not from standard physical chemistry, it should be derived or cited.

Choice of control through cycle rate (which affects compressor motor power use) is fine, but “off the shelf” is very limiting, and possibly paper could be edited to remove reference to this. Anyone using the algorithm would source a unit that just exceeds the predicted requirement anyway.

Equation 18: Log mean temperature difference does not work if phase and sensible heat changes occur simultaneously. You can see this from its derivation. No simple prediction equation exists for this situation, so you might as well use average temperature differences. In this case, if the refrigeration cycle runs under saturated conditions, then this is valid for the evaporator and not valid for the condenser.

Equation 19: remove dot over U.

Good work to include reasonably fundamental predictions of U (from h and k). I doubt such predictions would be accurate in practise (modelling flow through for example plate and tube heat exchanger is complex) but this is better than making up a number.

Equation 25 applies to air, and takes no account of product surface evaporation rates (through the vapour boundary layer adjacent to the product) or internal diffusion of moisture from within the product to the surface. This can only apply to a product in the constant rate period, a period which does not exist for mushroom drying (falling rate only). This error will lead to the program underestimating drying time.

Lines 196, 197: change “inferior” and “superior” to “less than” and “greater than”.

3. Results.

Logic of flowchart of algorithm appears to be correct. First decision box is confusing – does the algorithm stop if wetting occurs? Or only when ALL moisture is removed? In practise product is dried to its equilibrium moisture content with the air, not to zero, so the program will overestimate drying time.

What is significance of 0.0009 in algorithm? Possibly this is the product equilibrium moisture content, but this would limit the algorithm to the one product.

4. Discussion

Comparison with specific manufacturer’s design is probably not a good idea, unless its clearly explained to be a demonstration of how to use the algorithm to choose the specific design. Suggest clarifying this in discussion text (line 258 says this for the algorithm, but not for the refrigeration unit).

Line 287 – application of fuzzy logic would require a complete and extensive reworking of the problem and would be more work than developing the algorithm. I’m not sure this is a strong enough algorithm yet to warrant such development. Possibly after extensive testing and improvement with a range of products, this could be implemented.

5. Appendix A

Variables

Line 308: change dp to subscript.

Line 312: change cpm to cpm. Not sure why the ‘m’ is used – its not molar specific heat.

Line 318: Change U dot to simply U.

Reviewer 2 Report

Please add minimum several case studies as proof of the solution proposed.

Here a completion of my coments:   the method proposed is not new as heat pumps have specific already known technical utilization. One disadvantage is the energy source. it must be completed from renewable source. another disadvantage of the proposed three possibilities is that no examples are given, thus I recommend to make some case studies and indicate comparative results. Concerning the significance of content: medium, on my opinion not very novel ideas, only aplication of a known concept is adapted to a specific situation quality of presentation: good but can be improved by examples .
scientific soundness: good